# A Computationally Efficient Neuronal Model for Collision Detection with Contrast Polarity-Specific Feed-Forward Inhibition

**DOI:** 10.3390/biomimetics9110650

**Published:** 2024-10-22

**Authors:** Guangxuan Gao, Renyuan Liu, Mengying Wang, Qinbing Fu

**Affiliations:** Machine Life and Intelligence Research Centre, School of Mathematics and Information Science, Guangzhou University, Guangzhou 510006, China; 32116160276@e.gzhu.edu.cn (G.G.); rliu@e.gzhu.edu.cn (R.L.); 2112315082@e.gzhu.edu.cn (M.W.)

**Keywords:** collision detection, ON/OFF channels, feed-forward inhibition, computing efficiency, optimized LGMD, bio-robotics

## Abstract

Animals utilize their well-evolved dynamic vision systems to perceive and evade collision threats. Driven by biological research, bio-inspired models based on lobula giant movement detectors (LGMDs) address certain gaps in constructing artificial collision-detecting vision systems with robust selectivity, offering reliable, low-cost, and miniaturized collision sensors across various scenes. Recent progress in neuroscience has revealed the energetic advantages of dendritic arrangements presynaptic to the LGMDs, which receive contrast polarity-specific signals on separate dendritic fields. Specifically, feed-forward inhibitory inputs arise from parallel ON/OFF pathways interacting with excitation. However, none of the previous research has investigated the evolution of a computational LGMD model with feed-forward inhibition (FFI) separated by opposite polarity. This study fills this vacancy by presenting an optimized neuronal model where FFI is divided into ON/OFF channels, each with distinct synaptic connections. To align with the energy efficiency of biological systems, we introduce an activation function associated with neural computation of FFI and interactions between local excitation and lateral inhibition within ON/OFF channels, ignoring non-active signal processing. This approach significantly improves the time efficiency of the LGMD model, focusing only on substantial luminance changes in image streams. The proposed neuronal model not only accelerates visual processing in relatively stationary scenes but also maintains robust selectivity to ON/OFF-contrast looming stimuli. Additionally, it can suppress translational motion to a moderate extent. Comparative testing with state-of-the-art based on ON/OFF channels was conducted systematically using a range of visual stimuli, including indoor structured and complex outdoor scenes. The results demonstrated significant time savings in silico while retaining original collision selectivity. Furthermore, the optimized model was implemented in the embedded vision system of a micro-mobile robot, achieving the highest success ratio of collision avoidance at 97.51% while nearly halving the processing time compared with previous models. This highlights a robust and parsimonious collision-sensing mode that effectively addresses real-world challenges.

## 1. Introduction

With the advancement of science and technology, intelligent robots, driverless cars, drones, and other automated devices have increasingly become part of our daily lives. This progression brings the significant challenge of autonomous collision detection and avoidance [1,2]. The damage caused by collisions can be substantial, making effective collision detection crucial. To address this issue, various sensor approaches, such as radar [3], infrared [4], and laser [5], often combined with machine learning methods, as well as algorithms based on segmentation [6] and localization [7], have been widely employed in mobile machines. However, these sensor strategies face challenges in complex, dynamic environments regarding their reliability, scalability, and robustness to noisy signals. Deep learning-based methods have demonstrated superior accuracy in motion detection over the past decade. However, their computational cost is significantly high, particularly for visual processing.

Nature always surprises us with powerful sources. To achieve efficient and robust collision detection, researchers have increasingly focused on how animals address collision-detection challenges. Observations from nature have revealed that millions of years of evolution have equipped animals, particularly invertebrates, with powerful and efficient visual systems for sensing collisions. Different species employ various collision-detecting strategies and have distinct visual nervous system structures. For example, flies utilize fine placement of directional selective neurons [8], crabs rely on spatial localization vision ability [9], and bees depend on optic flow-based angular velocity estimation [10]. Researchers have extensively studied and modeled these biological systems, leading to the vibrant development of dynamic vision systems [1,11,12,13] and advanced sensor strategies [14,15].

Specifically, a pair of motion-sensitive sensors in the locust’s visual pathway, known as lobula giant movement detectors (LGMD1 and LGMD2), have been identified as looming motion detectors [16,17]. These LGMD neurons are located adjacent to each other and respond to the rapidly expanding image of an approaching object, indicating an imminent collision or predator strike. Despite their many commonalities, the neurons differ in their selectivity to collisions: LGMD2 is sensitive only to dark and vaguely appearing objects, whereas LGMD1 can detect both dark and light approaching stimuli.

Inspired by this fascinating ensemble of visual neurons, corresponding model systems for collision perception have been designed. The LGMD1 model is considered an approaching object detector, identifying potential collisions by responding to dilation near the edge of an object [18]. However, it is easily affected by irrelevant movements, such as translational and recessive motion, particularly in complex and dynamic environments. To enhance the performance, ON/OFF-channel-based LGMD models were proposed, including LGMD2 [19] and LGMD1 with ON and OFF pathways [20]. ON-contrast and OFF-contrast responses are processed in parallel within the ON/OFF channels, conveyed by interneurons of different polarities, resulting in specific direction selectivity (DS) for moving ON/OFF edges [21]. This has led to the establishment of an efficient and stable kind of collision detection system, shaped by biologically plausible pathways and mechanisms. However, it is important to note that there is still room for optimizing the computational resources of LGMD-based neural network models.

Recent physiological research consolidates the early findings that different types of inhibition are critically important in such looming sensitive neurons or counterpart model systems, including lateral inhibition and feed-forward inhibition (FFI), etc. [16,22,23,24]. Studies further investigated inhibitory signal processing presynaptic to LGMD [22,25]. The LGMD is synapsed with three dendritic fields labeled A, B, and C, as shown in Figure 1c. FFI from the OFF pathway acts in dendritic field C, while inhibition from the ON pathway acts in dendritic field B. Dendritic field C of LGMD presynaptic neurons receives stimulus inputs from transient inhibition via the OFF pathway in dorsal uncrossed bundle (DUB) axons, as well as ON-contrast excitatory stimulus. This suggests that inhibition is not unidirectionally integrated in the LGMD and that visual information does not efficiently form a retinotopic topological mapping in dendritic field C. In previous LGMD models, FFI is modeled numerically as average brightness change across the whole receptive field in a tiny time window regardless of contrast polarity [1]. Consequently, previous models ignored the placement of ON/OFF polarity in relation to FFI. How would the LGMD model evolve by modeling contrast polarity-specific FFI?

In the current modeling work, we aimed to incorporate this biological basis into the state-of-the-art LGMD model based on ON/OFF channels. Previously, FFI and lateral inhibition worked separately, with FFI acting across the lamina to respond to sudden changes in the visual scene, while lateral inhibition in the ON/OFF channels interacting with local contrast polarity-specific excitation. FFI can directly shut down the LGMD’s activities if the large area of luminance changes very rapidly. Such a computational role of FFI follows an all-or-none law that is not matching the latest biological findings. To make a difference, in our proposed model, inhibitory cells receive stimuli in the ON/OFF channels separately. To improve computational efficiency, we introduced an activation threshold function for the delivery of ON/OFF excitation and ignored inactive visual signals filtered by the second layer of the lamina. This approach significantly reduces computational resources in the subsequent neural network processing and aligns the model closer with the energy-saving principles of biological systems.

Accordingly, we propose an optimized LGMD (oLGMD) model with specific ON/OFF-contrast FFI. We conducted systematic experiments including offline stimuli tests and online robot tests to investigate the oLGMD with two state-of-the-art based on ON/OFF channels (i.e., LGMD1 [20] and LGMD2 [19]). The results demonstrated a few achievements that can be summarized as follows:This paper investigates polarity-specific FFI first from the perspective of computational modeling. Incorporating FFI into ON/OFF channels allows for the reception and processing of signals in parallel, consistent with the latest physiological findings. A simple and effective activation threshold function is proposed to the ON/OFF-contrast FFI, which addresses computational redundancy by ignoring stimuli unrelated to collisions in relatively stationary scenes, thereby improving the operational efficiency of the neural network. This approach also facilitates the moderate suppression of translational motion.The oLGMD model effectively implements diverse selectivity to either ON/OFF-contrast looming stimuli while retaining the original model’s robust performance, which would enrich the collision selectivity for vision-based mobile machines and offer a new candidate visual system for addressing real-world collision detection challenges.When applied to the embedded vision system of micro-robots, this model can be robust and energy efficient. The oLGMD model achieves the best performance while nearly halving the processing time compared with state-of-the-art.

Section 2 reviews the related research areas and applications. Section 3 presents the computational formulas and parameter settings of the proposed oLGMD model. Section 4 details the experimental setup, evaluation metrics, and result analysis, focusing on the advantages and uniqueness of ON/OFF FFI. Finally, Section 5 concludes the study and discusses future prospects. Notably, abbreviations used in this study and partial algorithms of the proposed methods and experiments are organized into a separate Appendix A accompanying this paper.

## 2. Related Work

In this section, we will discuss the most relevant research, including: (1) insect vision for collision detection, and (2) collision detection based on ON/OFF channels.

### 2.1. Insect Vision for Collision Detection

Understanding and modeling the visual systems of insects has captivated researchers for decades. Flying insects, known for their remarkable collision perception and avoidance abilities, have been the focus of numerous biological and modeling studies [1,11,12,21]. Extensive research has been conducted on flying insects using optical flow (OF)-based strategies [28,29,30], which offer computational simplicity and generate pixel-by-pixel motion flow to guide landing, collision avoidance, and navigational behavior. These strategies are currently applied to many collision systems for flying robots [11,12,31,32].

However, OF-based collision detection methods have several limitations. Firstly, they primarily perceive the threat of side collisions other than frontal collisions. Secondly, they exhibit weak detection capabilities for less textured, homogeneous objects. A newly found model employing this strategy is the lobula plate lobula column type-II (LPLC2) neuron model of flies [8,33,34,35,36]. LPLC2 neurons in flies cover all four stratified layers of the lobula plate tangential cells (LPTC), correlating responses in four cardinal directions to form ultra-selectivity to looming objects from the center of the receptive field. These neurons respond strongly to dark radial motion and more weakly to white radial motion, but neither dark nor white targets receding from the eye excite the LPLC2 neurons.

Moreover, biologists have anatomically explored a group of large interneurons in the lobula neuropile layer of the locust’s visual brain, known as lobula giant movement detectors (LGMDs) [37,38,39,40]. Figure 1a illustrates an LGMD neuron, showcasing its presynaptic and postsynaptic neural structures. This figure helps investigate the basic structure and mechanisms of the locust visual system to understand its signal processing scheme [23,41,42,43,44]. Neuromorphologically, LGMD1 integrates visual signals from different dendritic areas, generating two types of stimulations: excitatory and inhibitory. Neural processing within the circuit is a competition between these stimulations [45,46,47]. The descending contralateral movement detector (DCMD) is a one-to-one postsynaptic target neuron connected to the LGMD1 [48,49]. This neuron conveys the spikes generated by the LGMD1 to subsequent motion-control neural systems, producing avoidance behaviors [50]. Furthermore, LGMD2 is a neighboring partner to LGMD1, also serving as a looming detector. Although it shares similar characteristics, LGMD2 has different selectivity compared with LGMD1 [51]. Figure 1b illustrates the neuromorphological cross-sections of LGMD1 and LGMD2. Both neurons are physically adjacent to each other and have been functionally and anatomically identified in the lobula area. LGMD2 also has large fan-shaped dendrite. Importantly, physiological studies have shown the development of both neurons in locusts from adolescence to adulthood. LGMD2 matures earlier in juvenile locusts that lack wings and live mainly on the ground [27]. Consequently, LGMD2 plays a crucial role in juvenile locusts for perceiving predators from the sky.

### 2.2. Collision Detection Based on ON/OFF Channels

The ON and OFF pathway structures separate visual processing from the photoreceptor layer into parallel computation. These pathways have been identified in the preliminary visual systems of various animal species, including insects like flies [52] and vertebrates such as rabbits [53], mice [54], cats [55], and monkeys [56]. This structure underscores a fundamental principle of biological visual processing, where motion information is divided into parallel ON and OFF channels that encode luminance increments (ON) and luminance decrements (OFF) [57]. Specifically, the starting nerve cells of the ON channel favor dark-to-light luminance transitions (ON contrast), while the initiating nerve cells of the OFF channel respond to light-to-dark luminance transitions (OFF contrast).

The ON/OFF channels are crucial for DS in motion perception [21]. The computation of DS occurs separately within the ON and OFF pathways, culminating at the very next synapse where the DS is produced. Notably, the ON/OFF channels have been successfully integrated into robotic visual modules for real-time collision-detection tasks, demonstrating their efficacy and practicality in such applications [58].

Based on ON/OFF channels research, scholars hypothesized the existence of ON/OFF channels in locusts [21,59,60,61]. This led to modeling studies of two distinct LGMD neurons in the locust visual system, which are responsible for processing natural visual cues. LGMD1 neurons, possessing ON/OFF channels, have shown good performance in adverse situations, including low-contrast objects and highly textured backgrounds [20]. In the vicinity of LGMD1, LGMD2 was also computationally modeled via ON/OFF channels splitting preliminary motion into parallel computation with different delayed signal processing and non-linearity [19]. Differently from the LGMD1 neural network model, the LGMD2 model shows a unique preference for proximity of darker objects since ON channels are suppressed by stronger inhibitory signals in contrast to OFF channels. For the purpose of improving the performance in highly variable environments, adaptive inhibition mechanisms mediated by FFI in ON/OFF channels were proposed to act in conjunction with lateral inhibition [62]. Through evolutionary learning, this model outperformed previous methods to detect collisions in vehicle scenes. Over time, a number of computational models have also been developed to incorporate ON/OFF channels in LGMD models [21].

## 3. Methods and Materials

Within this section, we present a comprehensive description of the proposed oLGMD model with ON/OFF FFI (see Figure 2). The visual neural network is organized into two parts, namely feed-forward excitation (FFE) and FFI, both integrated at the LGMD. The model is designed to take video sequences as input, processing the frames in real-time to detect potential collisions. The inclusion of both excitation and inhibition within the ON/OFF channels allows for dynamic and adaptive responses to changes in the visual environment. Distinguishing itself from previous modeling approaches, our methodology introduces excitation and inhibition within the synaptic connections in the ON/OFF channels, aligning more closely with biological characteristics.

Specifically, FFE with ON/OFF channels handles the processing of visual stimuli by separating the incoming signals into ON and OFF channels, where ON channels encode luminance increments (brightening) and OFF channels encode luminance decrements (darkening). This separation allows for parallel processing of visual information, enhancing the model’s ability to accurately detect changes in the visual scene.

On the other hand, FFI with ON/OFF channels complements the excitation pathway by introducing inhibitory signals into the ON/OFF channels. The lateral inhibition mechanism is associated with FFI that suppresses irrelevant or non-collision-related signals, reducing noise and enhancing selectivity for potential collision events. By processing inhibitory signals in parallel with excitatory signals, the model achieves a more refined response to visual stimuli, closely mimicking biological processes.

This approach not only improves the model’s accuracy and efficiency in detecting collisions but also aligns with the energy-saving principles observed in biological visual systems. By incorporating biologically plausible pathways and mechanisms, the proposed neural network model offers a robust solution for real-time collision detection in complex and dynamic scenes.

As the neural computation of FFE is significantly consistent with the state of the art [19], the FFE algorithms are elaborated in the Appendix A, and only the new methods corresponding to ON/OFF-contrast FFI are formulated in this section. If the reader needs a comprehensive view of the neural network model, the Appendix A should be covered for better understanding.

### 3.1. Neural Computation of Feed-Forward Inhibition

Unlike all previous modeling studies, FFI is structured into parallel pathways relative to the DUB neurons in the medulla rather than directly inhibiting LGMD cells when there are rapid brightness changes across a large area of the visual field. Accordingly, the role of FFI is divided into two parts that interact with the FFE. The first part adjusts inhibitory signals during the linear computation of excitatory and inhibitory signals. The second part determines whether the incoming frame is processed by the network using the β-function in the FFE computation.

The computation of ON-contrast FFI obeys the following equations: (1)$$FFI_{on}\left(t\right)=\sum_{x=1}^{R}\sum_{y=1}^{C}P_{on}\left(x,y,t\right)\cdot {\left(C\cdot R\right)}^{-1}$$
where the mean ON-contrast response is taken from all ON transient cells $P_{on}$, and *C*, *R* denote the columns and rows of the visual field in a matrix form.
(2)$${\hat{FFI}}_{on}\left(t\right)=\alpha_{4}\cdot {FFI}_{on}\left(t\right)+\left(1-\alpha_{4}\right)\cdot {FFI}_{on}\left(t-1\right),\;\alpha_{4}=\tau_{in}/\left(\tau_{3}+\tau_{in}\right)$$
(3)$$\omega_{on}\left(t\right)=max\left(\omega_{1},\frac{{\hat{FFI}}_{on}\left(t\right)}{\omega_{ffi}}\right)$$
where $\tau_{3}$ denotes the delay time constant in milliseconds. $\omega_{ffi}$ is a predefined coefficient, and $\omega_{1}$ represents the baseline weight of the ON-contrast FFI. This configuration enhances lateral inhibition, making it more robust when there are significant luminance changes within the visual field.

Likewise, the computation of OFF-contrast FFI obeys the following equations: (4)$$FFI_{off}\left(t\right)=\sum_{x=1}^{R}\sum_{y=1}^{C}\left|P_{off}\left(x,y,t\right)\right|\cdot {\left(C\cdot R\right)}^{-1}$$
(5)$${\hat{FFI}}_{off}\left(t\right)=\alpha_{5}\cdot {FFI}_{off}\left(t\right)+\left(1-\alpha_{5}\right)\cdot {FFI}_{off}\left(t-1\right),\;\alpha_{5}=\tau_{in}/\left(\tau_{4}+\tau_{in}\right)$$
(6)$$\omega_{off}\left(t\right)=max\left(\omega_{2},\frac{{\hat{FFI}}_{off}\left(t\right)}{\omega_{ffi}}\right)$$
where $\omega_{2}$ represents the baseline weight of the OFF-contrast FFI.

Additionally, an activation threshold function is defined by the following formula to optimize the algorithm: if the ON or OFF contrast FFI value does not exceed the threshold, it indicates that the corresponded polarity channel is not highly activated by the external stimulus, resulting in the channel passing to the empty frame and not participating in further computation. Conversely, if the stimulus exceeds the threshold, it indicates that the polarity channel will participate in the following computation within ON/OFF channels. As a result, the β-function can be defined as
(7)$$\beta_{on}\left(t\right)=\left\{\begin{array}{cc} 0, & \mathrm{if}\;{\hat{FFI}}_{on}\left(t\right)\leq TH_{ffi}\\ 1, & \mathrm{if}\;{\hat{FFI}}_{on}\left(t\right)>TH_{ffi} \end{array}\right.$$
(8)$$\beta_{off}\left(t\right)=\left\{\begin{array}{cc} 0, & \mathrm{if}\;{\hat{FFI}}_{off}\left(t\right)\leq TH_{ffi}\\ 1, & \mathrm{if}\;{\hat{FFI}}_{off}\left(t\right)>TH_{ffi} \end{array}\right.$$

The β-function above is associated with the neural computation of local excitation as presented in the Appendix A, which can significantly accelerate the proposed algorithm, particularly in online robot visual processing. Specifically, if the ON/OFF-contrast FFI remains below a predefined threshold, the corresponding computation of the interaction between local excitation and lateral inhibition can be ignored.

### 3.2. LGMD Integration for Collision Detection

Similar to previous modeling studies, the LGMD neuron integrates all dendritic remaining excitation to form the FFE, which is then transformed into action potentials as the network output.
(9)$$FFE\left(t\right)=\sum_{x=1}^{R}\sum_{y=1}^{C}\hat{G}\left(x,y,t\right)$$
(10)$$K\left(t\right)={\left(1+e^{-FFE\left(t\right)\cdot {\left(C\cdot R\cdot \alpha_{6}\right)}^{-1}}\right)}^{-1}$$
where $\hat{G}\left(x,y,t\right)$ is the filtered local excitation from ON/OFF channels (see Appendix A). $\alpha_{6}$ denotes the scale factor, and the output is normalized into the range [0.5, 1). Subsequently, a spike frequency adaptation mechanism is applied to further enhance collision selectivity. The computation of this mechanism aligns with previous modeling work [20] and is thus omitted here. Membrane potentials are then mapped exponentially to firing rate.
(11)$$S^{spike}\left(t\right)=e^{\left(\alpha_{7}\cdot K\left(t\right)-T_{spi}\right)}$$
where $T_{spi}$ is the firing threshold and $\alpha_{7}$ is the scale parameter. A larger $\alpha_{7}$ corresponds to a higher firing rate. Finally, the following equation can be used to indicate a potential collision threat, particularly as an indicator for a robot’s collision avoidance.
(12)$$Col\left(t\right)=\left\{\begin{array}{cc} \mathrm{True}, & \mathrm{if}\;\sum_{i=t-n_{ts}}^{t}S^{spike}\left(i\right)\geq n_{sp}\\ \mathrm{False}, & \mathrm{otherwise} \end{array}\right.$$
where $n_{sp}$ denotes the number of spikes within a specified time window consisting of $n_{ts}$ consecutive digital signal frames.

### 3.3. Model Parameters

The parameter setting is given in the Appendix A. All parameters have been determined by optimizing the functionality of the proposed biologically viable pathways and mechanisms, as well as their suitability for an embedded vision system for micro-robots. The $TH_{ffi}$ parameter settings are discussed in detail in Section 4.

The algorithm described above can implement the specific selectivity of LGMD2, meaning it responds exclusively to OFF-contrast looming stimuli. By adjusting the spatiotemporal parameters in the ON/OFF channels, the proposed oLGMD model is capable of emulating both LGMD1 and LGMD2. For instance, by balancing the time delay and convolution weights in the ON and OFF pathways, the oLGMD model can replicate the functionality of LGMD1. The differing selectivities of these models will be demonstrated in Section 4.

### 3.4. Configuration

Firstly, the test environment for the offline experiment is configured with a 12th Gen Intel(R) Core(TM) i5-12500H processor, operating at a primary frequency of 2500 MHz and an outside frequency of 100 MHz. The system includes 16 GB of memory, 16 logical processors, and 12 cores.

Secondly, for online experiments, we introduce the micro-mobile robot *Colias* as illustrated in Figure 3. *Colias* is a low-cost, autonomous wheeled platform equipped with an RGB camera module. It has been successfully developed for bio-robotics research, including swarm robotics applications and neural vision systems research [63]. The detailed configuration related to this research and the robot’s online processing algorithm can be found in the Appendix A.

To evaluate the basic collision detection capabilities of the proposed framework and compare to related bio-inspired algorithms, an arena measuring 140 × 80 cm^2^ was built. The arena is enclosed by 15 cm high walls and illuminated uniformly from top to bottom. Top-down cameras were installed to track and record the micro-robot’s performance with overtime trajectories. As illustrated in Figure 3, the arena walls feature a distinct black and white grating pattern, and the *Colias* robot is equipped with ID-specific patterns on top to implement a real-time localization system [64]. This system assists us to record the robot’s trajectory and calculates the success ratio (SR) of collision detection.

## 4. Experimental Evaluation

This section details the systematic experiments conducted in this modeling study. To demonstrate the effectiveness of our model, we performed comparative experiments against the latest LGMD1 [20] and LGMD2 [19] neural network models, encompassing both offline and online tests. The offline tests involved visual stimuli from both computer simulations and real-world physical movements. Following these, we conducted online robot tests, including closed-loop arena competitions and open-loop tests, to assess computational time savings and the robustness of collision detection. Note that the proposed neural network framework can implement the specific collision selectivity of both LGMD1 (abbreviated as oLGMD1) and LGMD2 (abbreviated as oLGMD2) neuron models in the experiments.

### 4.1. Evaluation Criteria

To assess the model’s computational performance, we introduced a metric called the “efficiency ratio” (ER). This metric reflects the relationship between processing time and the proximity to a collision event. As a collision becomes more imminent, processing time increases, while in non-collision scenarios, the model remains idle, leading to shorter processing times. Consequently, a higher ER value signifies better model efficiency in effectively differentiating between collision and non-collision information. The ER is mathematically defined as follows:(13)$$ER=\frac{\sum_{i}^{F}\left(t_{max}-t_{i}\right)}{F\cdot t_{max}}$$
where $t_{max}$ represents the internal processing time required for the model to output the peak membrane potential at the moment of collision, and $t_{i}$ denotes the time at frame *i*. *F* is the total duration of the input image sequence. In the standard LGMD model, the computation time for processing each frame remains constant from the beginning of the run, regardless of whether a collision occurs. In contrast, the oLGMD model adjusts the computation time based on the stimulus intensity of each frame, as our proposed activation threshold function in ON/OFF-contrast FFI pathways can accelerate the visual processing against relatively stationary scenes. Accordingly, the ER metric can quantify the computing efficiency, indicating the time saved by the proposed model.

We also introduced a metric called the “success ratio” (SR) to evaluate the collision-detecting model’s accuracy. The SR metric provides a summary of the model’s performance by counting the number of correct and incorrect recognitions and categorizing them by class. Specifically, in the SR metric, true positives (TP) represent the number of collisions correctly identified by the model in collision scenarios. False positives (FP) indicate the number of instances where the model incorrectly identifies collisions in non-collision scenes. True negatives (TN) represent the number of non-collisions correctly identified by the model, while false negatives (FN) denote the instances where the model mistakenly identifies collisions in non-collision scenes. The SR is mathematically defined as follows:(14)$$SR=\frac{TP+TN}{TP+TN+FP+FN}$$

### 4.2. Synthetic Visual Stimuli Testing

To verify the basic functioning and effect of contrast polarity-specific FFI in the LGMD with ON/OFF channels, three sets of comparative experiments were conducted using synthetic stimuli, focusing on responses to approaching, receding, translating, and grating stimuli: (1) LGMD1 versus optimized LGMD1 (oLGMD1); (2) LGMD2 versus optimized LGMD2 (oLGMD2); and (3) oLGMD1 versus oLGMD2.

The experiments presented in Figure 4, Figure 5 and Figure 6 demonstrate that the proposed oLGMD model with contrast polarity-specific FFI retains its original collision detection capabilities for both approaching and receding movements. The oLGMD1 model performs consistently with the LGMD1 model [20], and the same results hold for oLGMD2 compared with LGMD2 [19]. Additionally, the proposed oLGMD model (whether oLGMD1 or oLGMD2) effectively suppresses translating stimuli to a moderate extent due to the newly introduced activation function in the ON/OFF FFI pathways, thereby enhancing the accuracy of collision detection. Furthermore, the model remains unresponsive to grating motion, just like the original models, maintaining consistent robustness against non-collision stimuli.

### 4.3. Real Physical Visual Movements Testing

To further evaluate the impact of the oLGMD, real-world stimuli tests were conducted using video recordings from both indoor structured scenes and vehicle crash scenarios. First, we assessed the robustness of the proposed model by presenting collision stimuli generated by the motion of a rolling ball in a structured indoor scene (Figure 7, Figure 8 and Figure 9).

The experiments reveal that the oLGMD model preserves the original collision-detection properties of LGMD1 and LGMD2 for dark objects eliciting ON/OFF contrast. Notably, the oLGMD effectively suppresses translating stimuli due to the activation threshold in the ON/OFF FFI pathways.

We also evaluated the efficacy of the proposed method in ground vehicle applications by using road recordings from dashboard cameras as visual stimuli (Figure 10 and Figure 11). In these scenes, the real-world environment presents additional background noise, such as flashing lights and shadows, which are common visual challenges for drivers. This setup provides a more rigorous test of the model’s collision detection capabilities.

Through experiments with real-world physical stimuli, we found that the oLGMD model (both oLGMD1 and oLGMD2) effectively ignores irrelevant stimuli, such as the movement of distant objects beyond a certain range and objects moving away from it. Conversely, it promptly triggers collision warnings and reacts strongly when objects approach from a specific angle and are about to collide. The model responds swiftly to potential collisions across various scenes in a stable and efficient manner, significantly enhancing computational efficiency (see statistical results in the next subsection). This performance is highly valuable for practical, real-world applications. Moreover, the results demonstrate that this biologically inspired structure has great potential to replicate the functional capabilities of biological LGMD1 and LGMD2 neurons.

### 4.4. Statistical Results

The above experiments demonstrated that the proposed oLGMD model is particularly effective in real-world noisy environments, successfully suppressing translational stimuli while retaining the original collision-sensitive capabilities. Moving forward, to corroborate the robustness and improved computing efficiency of the proposed method, we used a variety of visual stimuli to compare the models’ performance.

First, we measured the running time of the model across a variety of offline tests. This approach allows us to evaluate and portray the model’s computational efficiency in handling real-time collision detection. Figure 12 demonstrates the individual and integrated in-computer processing/running time of all comparative models. Our proposed oLGMD model (including oLGMD1 and oLGMD2) nearly halves the computing time across offline tests. On the other hand, how did the models perform in all collision events regarding the robustness?

We employed the SR metric in Equation (Equation 14) to evaluate the robustness of all comparative models. In this evaluation, a satisfactory output as TP or TN was scored as 1, while a failure as FP or FN was scored as 0. Table 1 illustrates the statistical results.

### 4.5. Investigation upon Collision Selectivity

During the above experiments, it was observed that the ON/OFF-contrast FFI implemented in the model could provide additional benefits beyond its primary contribution to improving computing efficiency.

Our research on the contrast polarity-specific FFI mechanism reveals that by adjusting the polarity delay parameters (τ), the scale parameter (ω), and the convolution kernel structure (*W*), the LGMD can be tuned to achieve various selectivity to ON/OFF contrast. This includes the development of an LGMD model with a new inhibition mechanism that exhibits selectivity opposite to that of the previous model. Additionally, this inhibition mechanism allows the model to suppress a temporary response that occurs during the initial phase of translational movements. Furthermore, the activation threshold enables the model to disregard minor or brief isolated excitation.

Figure 13 and Figure 14 demonstrate that we have implemented three types of selectivity through the proposed method: (1) ON/OFF contrast by responding to the approach of both darker and brighter objects (oLGMD1 model), (2) OFF contrast by responding to the approach of only darker objects (oLGMD2 model), and (3) ON contrast by responding to only brighter objects (reverse-oLGMD2 model).

### 4.6. Robot Online Testing

In the final phase of systematic experiments, we implemented the proposed model in the embedded vision system of a micro-mobile robot named *Colias* and compared the performance of oLGMD1 and oLGMD2 with state-of-the-art LGMD1 and LGMD2 models. We designed both closed-loop and open-loop robot experiments.

In the closed-loop experiments, we conducted arena tests where two model systems competed to measure the SR of collision detection and avoidance. Since motion control was outside the scope of this research, the robot was set to operate at a constant linear speed of approximately 0.04 m/s, with a near 180-degree turn upon detecting a potential collision. Three sets of trials were conducted, each lasting 30 min. Two robots were used in each test, interacting with both the arena walls and each other. Overtime trajectories of the robots were recorded using a real-time localization strategy as described in a related study [19].

In the open-loop experiments, we focused on the improvement in computational efficiency compared with related models through the robot’s embedded visual processing. Additionally, we explored the activation function to optimize the threshold parameter, aiming to enhance both the ER and SR.

Figure 15 presents the arena test results from three sets of robot competitions. Overall, all comparative models demonstrated competence in detecting collisions during navigation and dynamic interactions within the arena. The event density maps in Figure 16 highlight the performance of oLGMD1 and oLGMD2. In addition, Table 2 displays the SR for all comparative models. The results highlight two key findings: (1) the proposed method, incorporating contrast polarity-specific FFI, performs more robustly in arena tests, and (2) the oLGMD2 model outperforms all other models in rapidly sensing collisions in ground mobile robots, achieving a SR of 97.51%.

The computing efficiency, measured by time cost per defined time window, was compared through online processing of the robot in response to repeated collision stimuli. In these robot tests, the motion unit was deactivated. Figure 17 and Figure 18 clearly demonstrate the significant impact of optimization by comparing the running time of the original LGMD model with that of the oLGMD model. This achievement is evident on two levels: the increased efficiency observed across multiple collision scenarios and the enhanced operational efficiency during a single complete collision. The data demonstrate that the LGMD model maintains a fixed runtime during the collision scenario, whereas the oLGMD model can reduce the processing time by nearly half. The proposed model shows significant time savings when processing signals outside the collision time window, a result of the optimized ON/OFF FFI functions.

Finally, we investigated the key component of this model, specifically the activation threshold in the ON/OFF FFI pathways. Figure 19 presents the results using the robot’s embedded vision. Increasing the value of the activation threshold causes the model to ignore more visual information in frames. From the success ratio perspective, a smaller activation threshold allows the model to filter out some visual noise (such as translating), thereby increasing the SR. However, as the activation threshold continues to increase, more visual information is ignored, which inevitably leads to a decrease in SR. From the efficient ratio perspective, an increase in the activation threshold implies shorter computation time, leading to a rise in ER. We need to find a balance for the activation threshold under the constraints of these two indicators. As shown in Figure 19, when the activation threshold is between 0.75 and 1.75, both SR and ER are at high levels, meaning the model runs quickly while maintaining a high accuracy in collision detection.

In summary, the findings from these bio-robotic online tests suggest that optimizing insect vision through contrast polarity-specific feed-forward inhibition mechanisms offers a concise and effective solution for robotics. Recent biological studies continue to explore the application of ON/OFF mechanisms. To further enhance the inhibition mechanism, we propose using learning algorithms to fine-tune the thresholds for ON and OFF channels, thereby optimizing ${TH}_{ffi}$ within the ideal threshold range for specific light-dark scenarios. Additionally, improving robustness in collision detection is crucial for more challenging environments, such as diverse vehicle driving scenarios.

In essence, we anticipate that the visual system will exhibit its strongest response when a collision is imminent. During other times, such as during translational movements where the threshold is not exceeded, the system remains inactive to conserve energy. However, inhibition during translation operates within a specific range and is largely influenced by the threshold value. Thus, determining the appropriate threshold parameter is crucial, as it involves selecting an optimal value. Our ultimate goal is to ensure that the proposed model performs effectively under these conditions.

## 5. Conclusions

Throughout evolution, many organisms have developed specialized neural networks to effectively respond to approaching predators. Locusts in particular possess visual neurons known as LGMD neurons, which are highly sensitive to imminent collisions. Feed-forward inhibition (FFI) within LGMD neurons plays a crucial role in collision perception. Recent biological research has uncovered that presynaptic connections, specifically DUB neurons, contribute to the separation of FFI into ON/OFF contrasts.

Building on these findings, we proposed an initial implementation of FFI based on ON/OFF contrast. This mechanism, featuring a feed-forward neural network utilizing ON/OFF channels and energy-efficient thresholds, offers a simple yet effective method to reduce computation time in collision perception, especially in complex dynamic environments. The experiments demonstrated that the proposed oLGMD model, incorporating contrast polarity-specific FFI, achieves significant computational speedup while maintaining both selectivity and robustness.

Comparisons with previous models reveal that the oLGMD model effectively detects collisions in both physical and complex scenarios, retaining responsiveness and selectivity while significantly reducing processing time. This optimization highlights the potential of neuromorphic sensors for robotics and vehicular applications. As a result, the bio-inspired model has been optimized for time efficiency in real-world scenarios, including micro-robot implementation. Future work will explore additional applications of this mechanism, aiming to process more complex visual scenes to advance artificial vision and collision-detection systems.

## Figures and Tables

**Figure 1 biomimetics-09-00650-f001:**
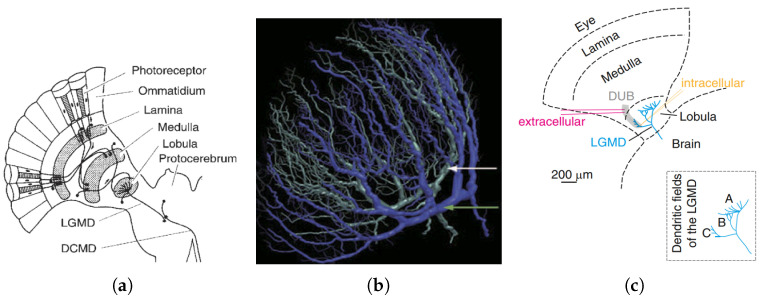
Neuromorphology of LGMD1 and LGMD2: (**a**) the presynaptic neuropile layers of the LGMD neuron and its postsynaptic one-to-one target, the DCMD neuron, image courtesy of [26]; (**b**) a 3D reconstruction of the dendritic trees of LGMD1 and LGMD2, indicated by white and green arrows, respectively, adapted from [27]; (**c**) the schematics of the locust’s optic lobe, where the inset illustrates the three dendritic fields of the LGMD, image courtesy of [22].

**Figure 2 biomimetics-09-00650-f002:**
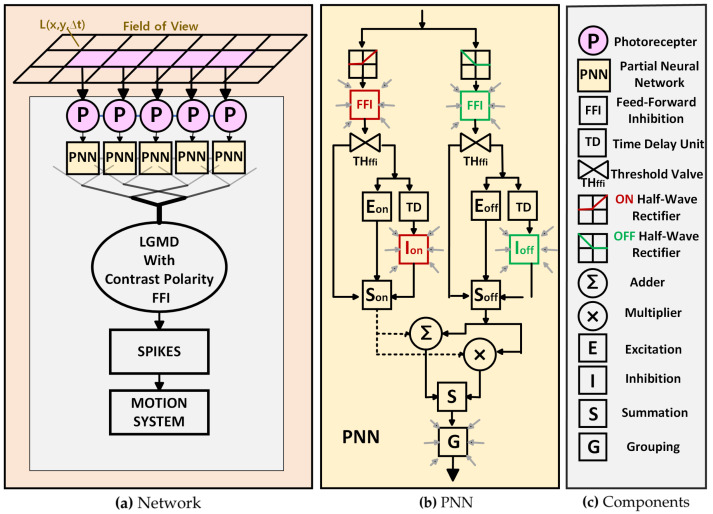
Illustrations of the optimized-LGMD neural network model: (**a**) the fundamental network structure of the model; (**b**) the detailed construction within each partial neural network (PNN); (**c**) the detail components.

**Figure 3 biomimetics-09-00650-f003:**
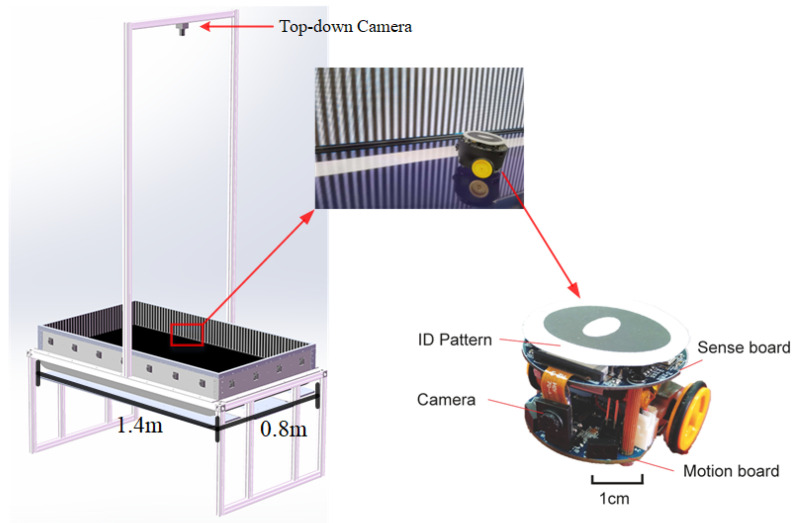
Illustration of *Colias* micro-mobile robot and the arena used in online experiments.

**Figure 4 biomimetics-09-00650-f004:**
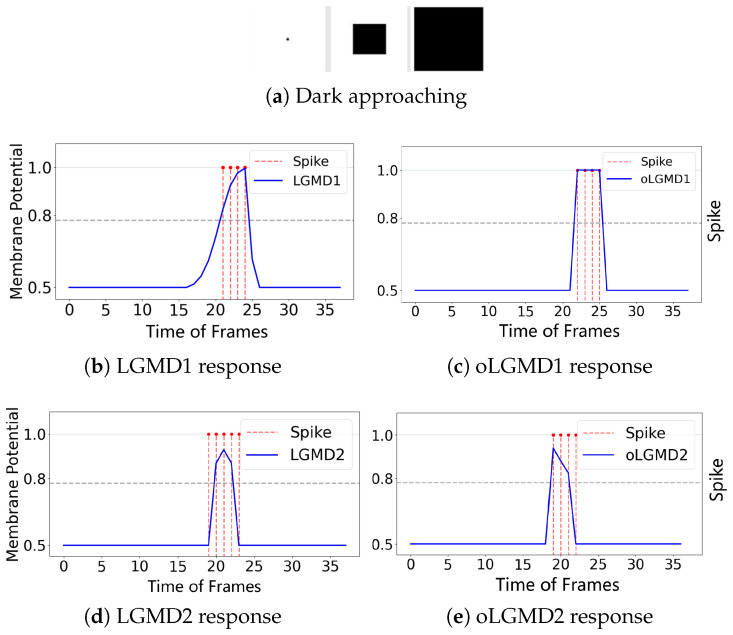
Illustrations of approaching stimulation and results: the horizontal axis represents the frames of the input video, while the vertical axis shows the membrane potential output of different models. When the membrane potential exceeds 0.8, the downstream neuron fires a spike, indicated by a red dashed line. (**a**) visual stimuli, (**b**) LGMD1’s membrane potential and spikes, (**c**) oLGMD1’s membrane potential and spikes, (**d**) LGMD2’s membrane potential and spikes, (**e**) oLGMD2’s membrane potential and spikes. The four comparative models all respond strongly to the proximity of object.

**Figure 5 biomimetics-09-00650-f005:**
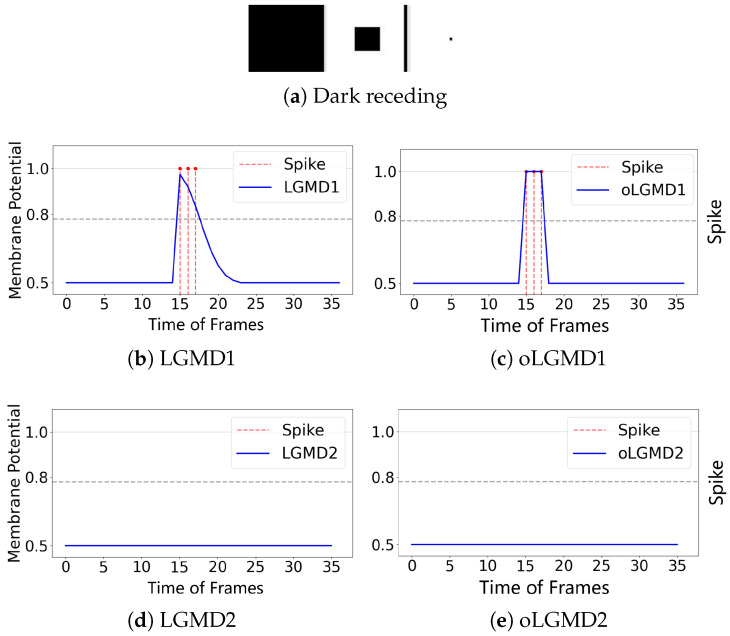
Illustrations of receding stimulation and results: LGMD2 and oLGMD2 models keep silent to dark receding, i.e., ON-contrast stimulation.

**Figure 6 biomimetics-09-00650-f006:**
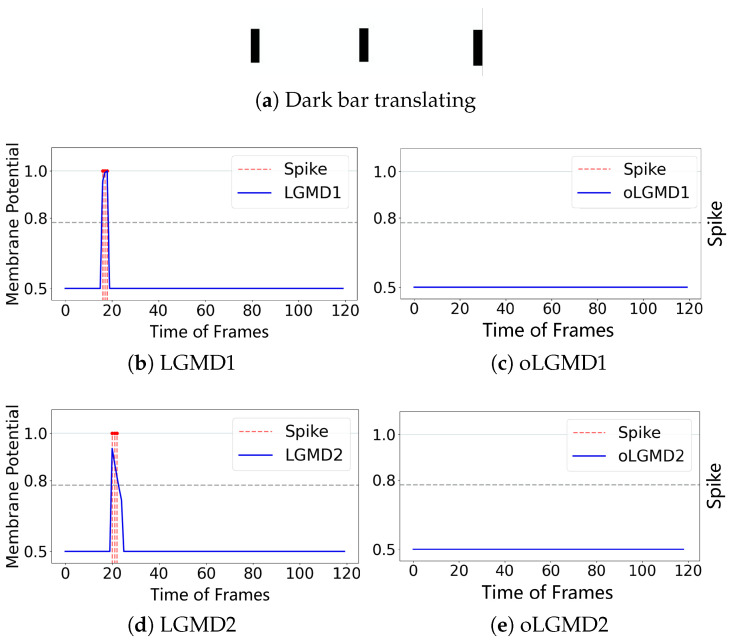
Illustrations of translating stimulation and results: oLGMD1 and oLGMD2 are inhibited against translating.

**Figure 7 biomimetics-09-00650-f007:**
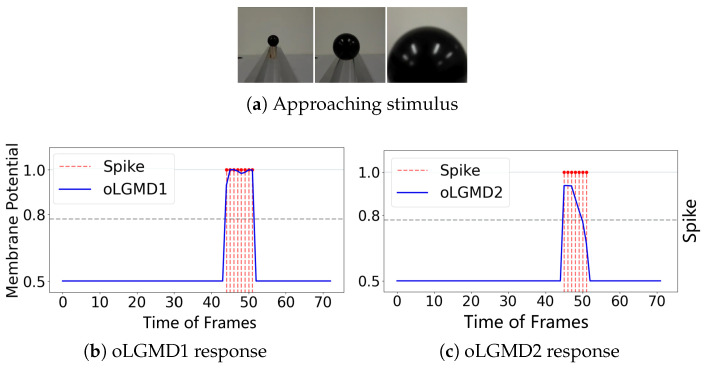
Illustrations of ball approaching test results of oLGMD model: both oLGMD1 and oLGMD2 respond strongly to the stimulation.

**Figure 8 biomimetics-09-00650-f008:**
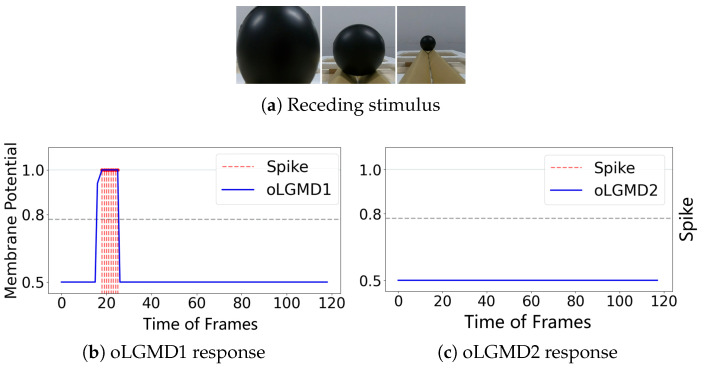
Illustrations of ball receding test results of oLGMD model: oLGMD1 responds briefly and oLGMD2 is unresponsive.

**Figure 9 biomimetics-09-00650-f009:**
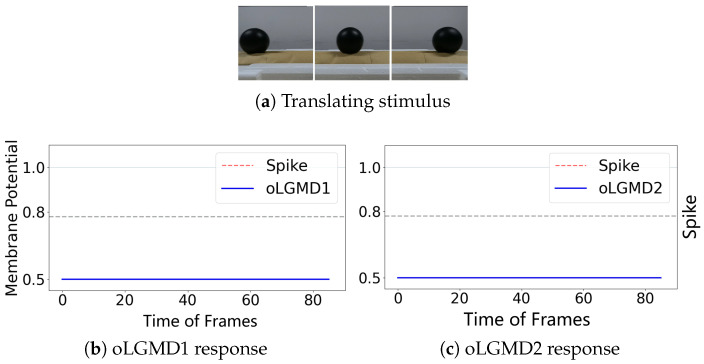
Illustrations of ball translating test results of oLGMD model: both oLGMD1 and oLGMD2 models well suppress excitation induced by translating.

**Figure 10 biomimetics-09-00650-f010:**
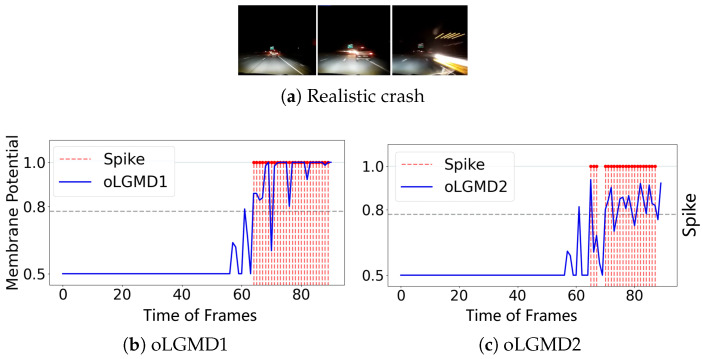
Results from the night crash tests demonstrate that both the oLGMD1 and oLGMD2 models can accurately predict imminent collisions under low-light conditions.

**Figure 11 biomimetics-09-00650-f011:**
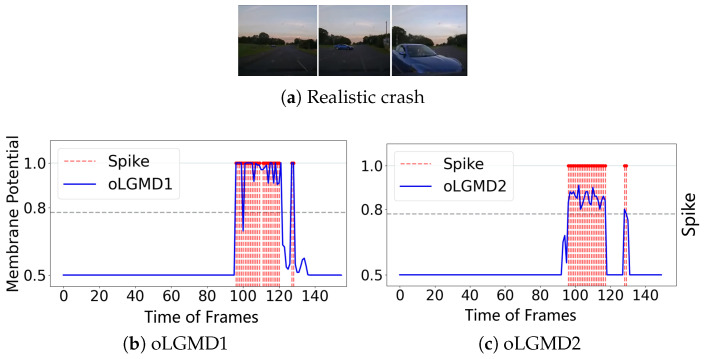
Daylight car crash tests illustrate that both the oLGMD1 and oLGMD2 models can precisely predict imminent collisions in bright, daytime conditions.

**Figure 12 biomimetics-09-00650-f012:**
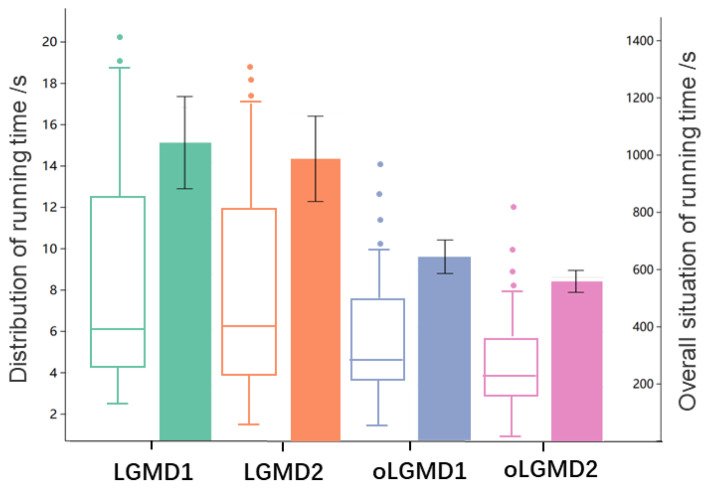
This figure presents data grouped into four sets corresponding to LGMD1, LGMD2, oLGMD1, and oLGMD2. Each is analyzed at two feature dimensions: (1) Box-and-whisker plot illustrates the distribution of the model’s required run times across all collision events of varying lengths. It highlights the individual characteristics of the run times, including the median, quartiles, and potential outliers. (2) Error bar graph displays the overall characteristics of the run times, with the error bars representing the range of variability across the experiments. The error bars indicate the uncertainty or variability in the model’s integrated processing times across all collision events. Both features represent data over a variety of experiments, providing a comprehensive view of the computational efficiency of each comparative model. Our proposed oLGMD model halves the processing time, showing a smaller variance.

**Figure 13 biomimetics-09-00650-f013:**
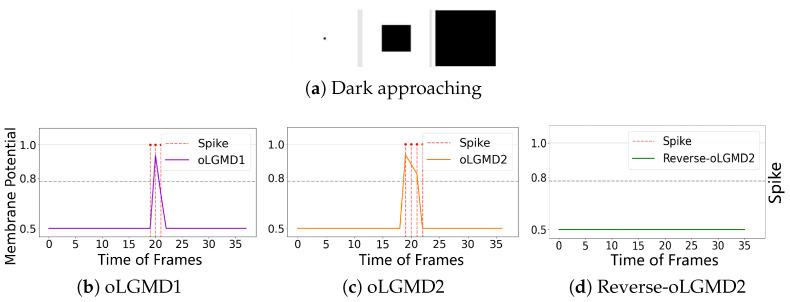
Illustrations of the responses of three collision selective models to dark approaching stimulation: the results indicate that both oLGMD1 and oLGMD2 generate collision warnings in response to the dark approaching stimulus, whereas reverse-oLGMD2 remains unresponsive.

**Figure 14 biomimetics-09-00650-f014:**
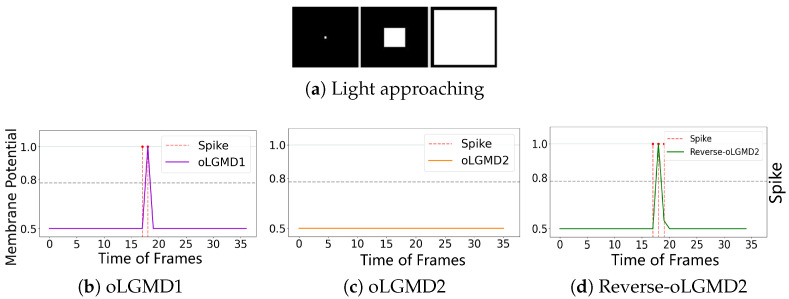
Illustrations of the responses of three collision selective models to light approaching stimulation: the results indicate that both oLGMD1 and reverse-oLGMD2 issue collision warnings in response to the light approaching stimulus, whereas oLGMD2 remains unresponsive. oLGMD2 and reverse-oLGMD2 models have the opposite looming selectivities.

**Figure 15 biomimetics-09-00650-f015:**
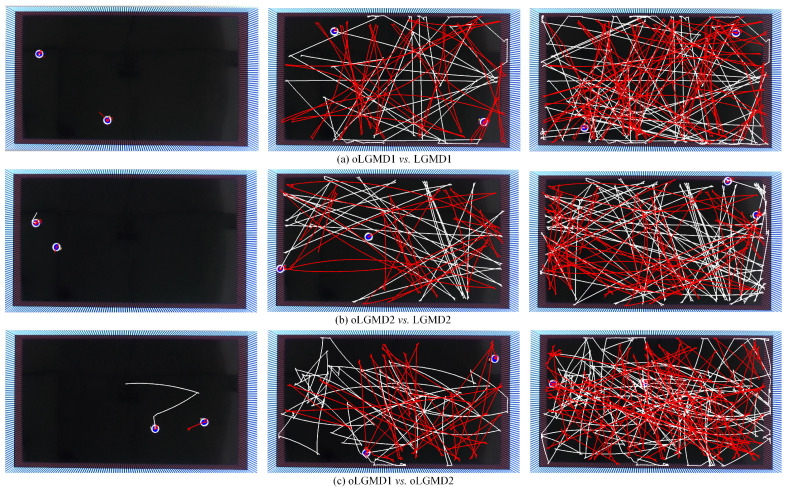
Illustrations of robot arena tests showing three processes with overtime trajectories: (**a**) oLGMD1 (white) vs. LGMD1 (red), (**b**) oLGMD2 (white) vs. LGMD2 (red), (**c**) oLGMD1 (white) vs. oLGMD2 (red). All *Colias* robots operated at a constant linear speed of approximately 0.04 m/s, with each process lasting 10 min.

**Figure 16 biomimetics-09-00650-f016:**
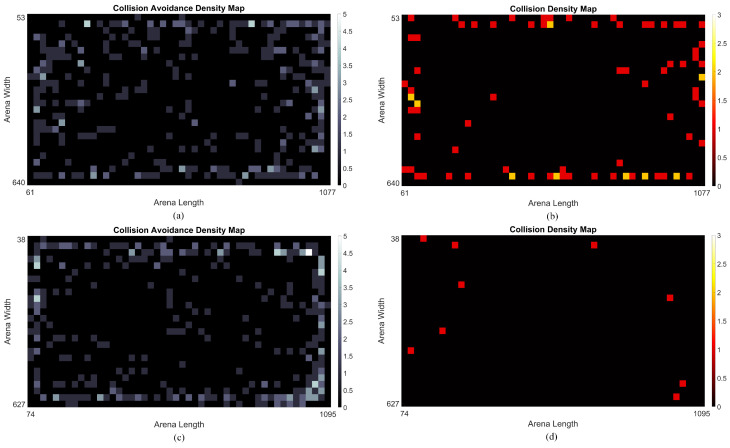
These density maps depict collision avoidance and crash events in the arena tests of the oLGMD model, with each model tested over a 30-min period. (**a**) oLGMD1 avoidance events, (**b**) oLGMD1 collision events, (**c**) oLGMD2 avoidance events, (**d**) oLGMD2 collision events.

**Figure 17 biomimetics-09-00650-f017:**
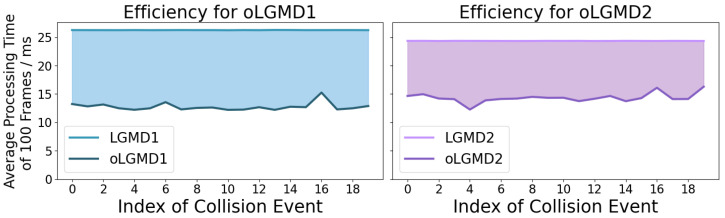
This figure compares the computational efficiency of LGMD1 versus oLGMD1 and LGMD2 versus oLGMD2 in response to 20 collision stimuli. In each scenario, the micro-robot processes 100 frames of raw images, with processing times averaged over these frames in milliseconds, as measured by a real-world clock system. The shaded area highlights the significant improvement in computational efficiency achieved by the oLGMD model compared with the LGMD model across these collision scenarios.

**Figure 18 biomimetics-09-00650-f018:**
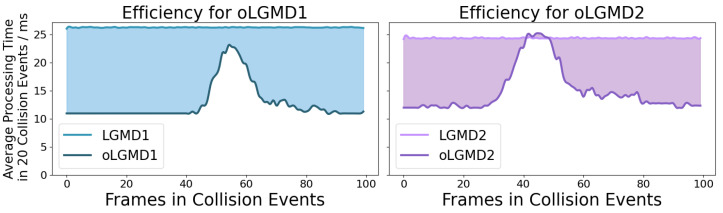
This figure compares the average time costs of oLGMD1 versus LGMD1 and oLGMD2 versus LGMD2 in response to 20 collision stimuli. Each visual movement consists of 100 frames, and the curves represent the average time measured across 20 repeated tests. The shaded area highlights the substantial improvement in computational efficiency of the oLGMD models. The oLGMD model significantly reduces processing time outside the collision time window, achieved through the ON/OFF FFI pathways.

**Figure 19 biomimetics-09-00650-f019:**
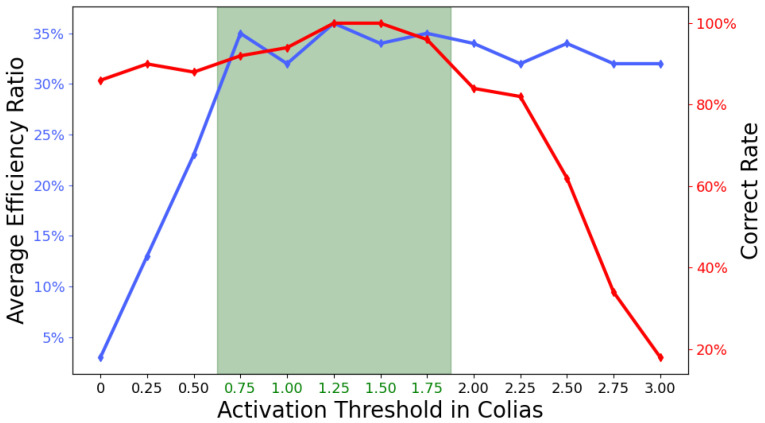
This figure illustrates the robot’s performance with varying activation thresholds in the ON/OFF FFI pathways by examining the computing efficiency ratio (ER, Equation (Equation 13)) and the success ratio (SR, Equation (Equation 14)) of collision detection. The horizontal axis represents different activation thresholds. The left vertical axis shows the average operational ER of the proposed model when subjected to a collision scenario to stimulate the *Colias* vision system. The right vertical axis indicates the collision detection SR under the same conditions. The collision process was repeated 50 times to calculate the average ER and SR. As the activation threshold increases, the oLGMD model processes progressively less visual information, leading to higher computational efficiency but a lower collision recognition rate. The green-shaded area represents the range of activation threshold values that effectively balances these two factors.

**Table 1 biomimetics-09-00650-t001:** Success ratio of the comparative models across offline tests.

LGMD1	LGMD2	oLGMD1	oLGMD2
79 ± 5%	91 ± 3%	86 ± 2%	97 ± 1%

**Table 2 biomimetics-09-00650-t002:** Success ratio of arena tests.

Model	Events (Collisions)	SR
LGMD1	437 (84)	80.78%
oLGMD1	414 (64)	84.54%
LGMD2	381 (10)	97.38%
oLGMD2	362 (9)	97.51%

## Data Availability

The raw data supporting the conclusions of this article will be made available by the authors on request. Source code can be accessed at https://github.com/Ryannnice/Supplementary_Materials_FFI_ON_OFF (accessed on 16 October 2024).

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
