# Peer review of "A Computationally Efficient Neuronal Model for Collision Detection with Contrast Polarity-Specific Feed-Forward Inhibition"

_biomimetics, 2024, doi:10.3390/biomimetics9110650_

Round 1
Reviewer 1 Report
Comments and Suggestions for Authors
This paper reports a Computationally Efficient Collision-Detection Neuronal Model with Contrast Polarity-Specific Feed-Forward Inhibition. The paper proposes a neuronal model that accelerates 15 visual processing in relatively stationary scenes and also maintains robust selectivity to ON/OFF- 16 contrast looming stimuli. The optimized model was implemented in the embedded vision system of a micro-mobile robot, achieving the highest success ratio of collision avoidance at 97.51%. The paper is interesting and can be potentially utilized in improving the vision of intelligent robots. I have the following comments/concerns about the submitted manuscript that could improve its quality and readability.
1. The authors need to provide the main contribution and novelty of their proposed technique over the already reported in the literature. There are a lot of techniques already reported in the field of improvement of vision for intelligent robots. The authors should comment on why the proposed model is better than the existing solutions.
2. The title of the paper should be corrected as “A computationally efficient” and not “A computing efficient”
3. The authors claimed that computational modeling of contrast polarity-specific FFI that includes the addition of an activation threshold to the inhibition mechanism. How does this approach address computational redundancy if one ignores the stimuli not related to collisions in stationary scenarios? And how do they improve the operational efficiency of the neural network?
4. The authors claim that their proposed model is effective to implement diverse selectivity to ON-OFF looming stimuli while still retaining the performance of the model? The authors should comment if this approach going to ensure collision selectivity for vision-based robots?
5. The authors are claiming that using FFI into ON/OFF channels allow for processing of signals. However, when they are applied to the embedded vision system of micro-robots, this model can be robust and energy efficient. The authors should provide evidence for their claim by mentioning how much energy is saved during this process, as energy efficiency is a big concern when considering the long-term usage of these robots.
6. The authors should also comment on how the parameter settings of the proposed oLGMD model were optimized. The whole process should be elaborated to support the claim.
7. In Figure 2, it is better to label them as (a), (b), and (c).
8. Including Figure 4 and onwards, the x-axis and y-axis are not mentioned.
9. In Fig. 20, the authors should explain, why the range from 0.75 to 1.75 has so much significance,
1. There are a few typos in the manuscript that need to be fixed. For example, line 255, should be “which will be introduced later” and not “which will introduced later”, In line 151, the word “tis” should be replaced with “its”
Comments on the Quality of English LanguageThere are a few typos in the manuscript that need to be fixed. For example, line 255, should be “which will be introduced later” and not “which will introduced later”, In line 151, the word “tis” should be replaced with “its”
Author Response
Comments 1: [This paper reports a Computationally Efficient Collision-Detection Neuronal Model with Contrast Polarity-Specific Feed-Forward Inhibition. The paper proposes a neuronal model that accelerates 15 visual processing in relatively stationary scenes and also maintains robust selectivity to ON/OFF- 16 contrast looming stimuli. The optimized model was implemented in the embedded vision system of a micro-mobile robot, achieving the highest success ratio of collision avoidance at 97.51%. The paper is interesting and can be potentially utilized in improving the vision of intelligent robots. I have the following comments/concerns about the submitted manuscript that could improve its quality and readability.]
Response: Thanks for your valuable time and feedback on this research paper. We appreciated every recommendations and made substantial changes on the revised paper. The novelty and contribution of this paper currently would be easier to caught by readers.
Comments 2: [The authors need to provide the main contribution and novelty of their proposed technique over the already reported in the literature. There are a lot of techniques already reported in the field of improvement of vision for intelligent robots. The authors should comment on why the proposed model is better than the existing solutions.]
Response: Thanks for this comment. We have summarized the main contributions and novelties in the Introduction. Moreover, it would be easy for readers to grab the main innovation of this paper in every sections. The proposed collision perception method retains original robustness and diverse selectivity, but is more computing efficient due to the separation of ON/OFF contrast FFI and the activation threshold function. The robot performance was even better than the state of the art.
Comments 3: [The title of the paper should be corrected as “A computationally efficient” and not “A computing efficient”]
Response: Thanks for the detailed suggestion. We have corrected the title.
Comments 4: [The authors claimed that computational modeling of contrast polarity-specific FFI that includes the addition of an activation threshold to the inhibition mechanism. How does this approach address computational redundancy if one ignores the stimuli not related to collisions in stationary scenarios? And how do they improve the operational efficiency of the neural network?]
Response: Thanks for this constructive comment. The activation threshold function in ON/OFF FFI pathways is critical to speed up the neural network processing. As the FFI indeed reflects the whole-field luminance change within a tiny time window. In relatively stationary scenes, there would be distracted motion that cannot activate the FFI pathways. Accordingly, the ON/OFF channels can be idle in this case. We have emphasized these points in the revised paper.
Comments 5: [The authors claim that their proposed model is effective to implement diverse selectivity to ON-OFF looming stimuli while still retaining the performance of the model? The authors should comment if this approach going to ensure collision selectivity for vision-based robots?]
Response: Thanks for this comment. We thoroughly investigated the ON/OFF-contrast looming stimuli in offline tests to verify the proposed method can implement diverse selectivity. As the foundation of this model originates from the state of the art LGMD1 and LGMD2 models, we mainly focused on investigating its robustness and computing efficiency as embedded vision systems in robot experiments. The different selectivities were demonstrated previously in robots.
Comments 6: [The authors are claiming that using FFI into ON/OFF channels allow for processing of signals. However, when they are applied to the embedded vision system of micro-robots, this model can be robust and energy efficient. The authors should provide evidence for their claim by mentioning how much energy is saved during this process, as energy efficiency is a big concern when considering the long-term usage of these robots.]
Response: Thanks for this comment. We collected computing time cost from the internal clock system of embedded vision systems and found the proposed method nearly halved the processing time due to the separation of FFI. The results were shown in our experiments (Figure 17 and 18).
Comments 7: [The authors should also comment on how the parameter settings of the proposed oLGMD model were optimized. The whole process should be elaborated to support the claim.]
Response: We agree that the parameters setting is always influencing the model performance when the testing environments are changing. Therefore, we adapted the parameters from both biological and computation modeling from previous experience. The key parameters were chosen carefully in previous studies and the new parameters were also investigated systematically in the proposed research. The proposed model works effectively across various scenarios from synthetic to robot.
Comments 8: [In Figure 2, it is better to label them as (a), (b), and (c).]
Response: We corrected this Figure to make it clear and consistent.
Comments 9: [Including Figure 4 and onwards, the x-axis and y-axis are not mentioned.]
Response: We unified the x-axis and y-axis of the same category of Figures.
Comments 10: [In Fig. 20, the authors should explain, why the range from 0.75 to 1.75 has so much significance]
Response: Thanks for your careful reading. We highlighted our results and explained the influence of activation threshold in the caption of Figure 20, as well as in the text of Experiments Section.
Comments 11: [There are a few typos in the manuscript that need to be fixed. For example, line 255, should be “which will be introduced later” and not “which will introduced later”, In line 151, the word “tis” should be replaced with “its".]
Response: Thanks for your rigorous reading. We corrected these typos, and proofread this paper.

Reviewer 2 Report
Comments and Suggestions for Authors
The authors present a well founded approach to build bio-inspired time-to-contact detectors.
The proposed approach takes inspiration from several biological evidences and propose well conceived solutions to implement them in neuromorphic circuits.
The Mathematical modeling is convincing.
As a whole, the work strongly grounds on several works previously published by the same authors (eight of them were referenced). This is the major concern with the paper, since most of the material has been already published and the claimed novelty is slightly incremental (see in particular the recent review Fu, Q. Motion perception based on ON/OFF channels: A survey. Neural Networks 2023, 165, 1–18.).
The proposed architecture, compared with previous ones (see also the enclosed figure), includes additional front-end feed-forward inhibition (FFI). This extension improves the model’s accuracy and efficiency in detecting collisions, while aligning with the energy-saving principles observed in biological visual systems.
From this perspective, the proposed manuscript is not straightforwardly accessible and largely redundant with previous studies. On my point of view, the paper should be revised as a "Technical note" or "Brief article", i.e., considerably shortened by summarizing the key features of the previous architecture, and highlighting the major differences or add-on's (i.e., the FFI stage). The goal should be to make the underlying principles immediately graspable, and to let them to be appreciated, by introducing the rationale and the expected advantages of the proposed modification. Results should better focus on the advantages with respect to the non-optimized network: i.e., (1) on the the responses to translating stimuli, effectively suppressed due to the activation threshold in the ON/OFF FFI pathways, and (2) on energy saving issues. Other minor results (see Fig.7 just an an example) could be discussed in the last section ("Discussion and Conclusion"), which is presently quite skimpy, while they can be fully reported as Supplementary Material.

Author Response
Comments: [The authors present a well founded approach to build bio-inspired time-to-contact detectors. The proposed approach takes inspiration from several biological evidences and propose well conceived solutions to implement them in neuromorphic circuits. The Mathematical modeling is convincing. As a whole, the work strongly grounds on several works previously published by the same authors (eight of them were referenced). This is the major concern with the paper, since most of the material has been already published and the claimed novelty is slightly incremental (see in particular the recent review Fu, Q. Motion perception based on ON/OFF channels: A survey. Neural Networks 2023, 165, 1–18.). The proposed architecture, compared with previous ones (see also the enclosed figure), includes additional front-end feed-forward inhibition (FFI). This extension improves the model’s accuracy and efficiency in detecting collisions, while aligning with the energy-saving principles observed in biological visual systems. From this perspective, the proposed manuscript is not straightforwardly accessible and largely redundant with previous studies. On my point of view, the paper should be revised as a "Technical note" or "Brief article", i.e., considerably shortened by summarizing the key features of the previous architecture, and highlighting the major differences or add-on's (i.e., the FFI stage). The goal should be to make the underlying principles immediately graspable, and to let them to be appreciated, by introducing the rationale and the expected advantages of the proposed modification. Results should better focus on the advantages with respect to the non-optimized network: i.e., (1) on the the responses to translating stimuli, effectively suppressed due to the activation threshold in the ON/OFF FFI pathways, and (2) on energy saving issues. Other minor results (see Fig.7 just an an example) could be discussed in the last section ("Discussion and Conclusion"), which is presently quite skimpy, while they can be fully reported as Supplementary Material.]
Response: Many thanks for your valuable time and constructive feedback on this research paper. We appreciated your points of view on the main novelty and contribution, also the length of this paper. Here we respond these as a whole. According to your recommendations, we have made substantial changes of this paper as the following aspects:
- We organized the neural computation of feed-forward excitation into the Supplemental File (attached for your convenience), as most algorithms are consistent with the state of the art, our previous modeling studies. As a result, the new formulation of the ON/OFF contrast specific FFI can be easily spotted at the beginning of Method Section. This paper is much shortened from 26 to 21 pages of manuscript format.
- We moved the tables of nomenclature, parameters into the Supplemental File. The online algorithm of robot implementation was also integrated into the Supplemental File.
- We moved the introduction of applied micro-robot into the Supplemental File to make the Configuration of main text more concisely. The corresponding references are also given in the Supplement.
- We only moved the grating tests results into the Supplement. We rigorously checked other experimental results and found they are all important to demonstrate our main achievements, gradually and clearly.
- We further highlighted our main contributions compared to our previous modeling works at every Sections. The main novelty is the separation of FFI into ON/OFF-contrast and the activation threshold function that significantly speed up the algorithm as the visual processing of relatively stationary scenes will be omitted by the ON/OFF channels, and only the significant change like the expanding edge near the end of approaching would be extracted and processed by the medulla and lobula layers.
- We enriched the references (64) and reduced self-citations (8) by keeping only the most necessary and related ones.
- We proofread and corrected English writing and this paper would be more readable.

Round 2
Reviewer 1 Report
Comments and Suggestions for Authors
The authors' have made the suggested corrections.
Reviewer 2 Report
Comments and Suggestions for Authors
Highlighting the changes in the manuscript would have been appreciable.
Anyway, the revision made by the authors substantially answer most of my concerns.